

# Seed traits and phylogeny explain plant's geographic distributions

Kai Chen[1,2,7], Kevin S. Burgess[3], Fangliang He[4], Xiang-Yun Yang[1], Lian-Ming Gao[2,5], De-Zhu Li[1,2,6]

[1]Germplasm Bank of Wild Species in Southwest China, Kunming Institute of Botany, Chinese Academy
of Sciences, Kunming, Yunnan 650201, China

[2]CAS Key Laboratory for Plant Diversity and Biogeography of East Asia, Kunming Institute of Botany,
Chinese Academy of Sciences, Kunming, Yunnan 650201, China

[3]Department of Biology, College of Letters and Sciences, Columbus State University, University
System of Georgia, Columbus, GA 31907-5645, USA

[4]Department of Renewable Resources, University of Alberta, Alberta, Canada

[5]Lijiang Forest Biodiversity National Observation and Research Station, Kunming Institute of Botany,
Chinese Academy of Sciences, Lijiang 674100, Yunnan, China

[6]Kunming College of Life Science, University of Chinese Academy of Sciences, Kunming, Yunnan
650201, China

[7] Key Laboratory of Insect Resources Conservation and Utilization in Western Yunnan, Baoshan
University, Baoshan, Yunnan 678000, China

Running title: Seed traits and phylogeny explain plant distribution

Corresponding authors: Lian-Ming Gao (gaolm@mail.kib.ac.cn), De-Zhu Li (dzl@mail.kib.ac.cn)





**Abstract.** Understanding the mechanisms that shape the geographic distribution of plant species is a
central theme of biogeography. Although seed mass, seed dispersal mode and phylogeny have long been
suspected to affect species distribution, the link between the sources of variation of these attributes and
their effects to the distribution of seed plants are poorly documented. This study aims to quantify the
joint effects of key seed traits and phylogeny on species' distribution. We collected seed mass and seed
dispersal mode from 1,426 species of seed plants representing 501 genera of 122 families and used
4,138,851 specimens to model species distributional range size. Phylogenetic generalized least squares
regression and variation partitioning were performed to estimate the effects of seed mass, seed dispersal
mode and phylogeny on species distribution. We found that species distributional range size was
significantly constrained by phylogeny. Seed mass and its intraspecific variation were also important in
limiting species distribution, but their effects were different among species with different dispersal
modes. Variation partitioning revealed that seed mass, seed mass variability, seed dispersal mode and
phylogeny together explained 46.82% of the variance in species range size. Although seed traits are not
typically used to model the geographic distributions of seed plants, our study provides direct evidence
showing seed mass, seed dispersal mode and phylogeny are important in explaining species geographic
distribution. This finding underscores the necessity to include seed traits and the phylogenetic history of
species in climate-based niche models for predicting the response of plant geographic distribution to
climate change.

**Keywords.** dispersal mode, distributional range size, phylogeny, seed mass, seed mass variability



# 1 Introduction

Understanding the ecological and evolutionary processes that govern the geographic range of species
can provide insights into their potential adaptive response to global climate change (Gaston and Fuller,
2009; Kubota et al., 2018). It is well known that the geographic ranges of species can span 12 orders of
magnitude, and closely related species may vary enormously in their range (Brown et al., 1996). Many
factors contribute to this variation, although dispersal ability and energy requirements associated with
establishment and persistence in varying habitats have been considered to be the two most important
ones (Morin and Chuine, 2006; Zhou et al., 2021). Given that seeds are the predominately mobile stage
of sessile plants, and seed mass generally reflects the amount of energy that a seed contains and its
mobility (Coomes and Grubb, 2003), it seems likely that seed mass could play an important role in
governing the geographic ranges of seed plants.

Seed mass can influence the colonization and competition ability of plant species along different
environmental gradients (Chen et al., 2018; Bu et al., 2019). Large-seeded species more often occupy
habitats that have high levels of energy (i.e., tropical or low elevation habitats) and tend to be better
competitors in these environments (Moles and Westoby, 2004), where they typically have higher
germination rates (Akaffou et al., 2021), and greater seedling survivorship (Mukherjee et al., 2019).
Small-seeded species, however, usually occupy low energy habitats. They often produce a large amount
of seeds, allowing them to arrive in new (possibly harsher) habitats through wind dispersal (Greene and
Quesada, 2005; Morin and Chuine, 2006; Sonkoly et al., 2017). Furthermore, seed mass has been shown
to decrease along increasing environmental extremes, indicative of the superior colonization ability of
small-seeded species in low energy habitats compared to that of large-seeded species (Procheș et al.,
2012; DeMalach et al., 2019). While some studies (e.g., Morin and Chuine 2006; Procheș et al., 2012)
indicate that species with small and light seeds tend to possess large geographic ranges, there is a need





to further quantify the relationship between seed mass and distributional range size across a broader suite of species and at a wider spatial scale.

Seed traits, including seed mass, could also vary considerably within species, which may be driven by plasticity genes or even the entire genome (Nicotra et al., 2010). Therefore, intraspecific seed mass
variation reflecting a species' high genetic diversity can enable adaptive response to varying environmental conditions and changing climate (Cochrane et al., 2015; Yang et al., 2016), so that to occupy more local habitats (Silvertown, 1989; Sides and Sloat, 2014). Although intraspecific seed mass variation could be an important factor influencing the geographic distribution of plants, few studies have evaluated this source of variation in a regional context.

The seed dispersal mode of a particular species, a key trait responsible for dispersal distance, can also greatly influence species geographic range (Oakwood et al., 1993; Chen et al., 2019b). The seed dispersal ability of a plant species is often a trade-off with other life-history characteristics, such as seed mass, morphologies and persistence in the soil, which in turn can affect seed germination, and the survival and growth of seedlings (Nathan, 2001; Chen and Valone, 2017). However, little is known
about the effect of dispersal modes on species distribution. It is also because of the tradeoff between dispersal modes and seed mass variation (Moles et al., 2007; Chen et al., 2019a), discerning the relative importance of seed mass and dispersal on the geographic distribution of seed plants is important but elusive.

Because species from a common ancestor typically experience similar selection pressures in
similar habitats, e.g., adaptive niche convergence (Losos, 2008; Grossenbacher et al., 2015), the geographic distribution of species is likely correlated in phylogenetic relationships. Furthermore, phylogenetic relatedness could also influence other ecological processes such as niche partitioning in overlapping habitats or variation in life-history traits, seed traits included, which in turn may influence the distribution range size of species (Moles et al., 2005). Therefore, a species' age or the degree of





phylogenetic relatedness could invoke biogeographic limits to expansion (Martin and Husband, 2009) or promote the evolutionary divergence of species and the variation in seed traits (Donoghue et al., 2001; Moles et al., 2005). Although a species' geographic range could well be dependent on its evolutionary history (Felsenstein, 1985), few studies have included phylogeny to discern the effect of seed traits on species distribution.

100        In this study, we attempted to quantify the effects of seed mass, intraspecific seed mass variation, dispersal mode and phylogeny on species geographic range size. We hypothesized that species possessing small seeds with high variability in seed mass, coupled with a strong dispersal capacity, would have larger distributional range sizes than species with contrasting seed traits, and furthermore, species distribution range is phylogenetically conserved. We collected data on seed mass and seed

dispersal mode from 1,426 plant species distributed mainly across China. We specifically aimed to answer two questions: (1) What are the joint effects of seed mass, seed dispersal and phylogeny on species geographic range size? and (2) Are there significant phylogenetic signals associated with species geographic range size?

## 2 Materials and methods

### 2.1 Seed mass data

Our dataset contains seeds of 1,426 species, representing 501 genera and 122 families of seed plants. All species occur in China, of which about 30% are endemic to China. Seeds from two to 136 populations for each of the species (a total of 17,223 populations) were obtained from the Germplasm Bank of Wild Species in Southwest China (GBOWS: http://www.genobank.org/). In addition, 549 populations for 454

of the 1,426 species (one to six populations per species) were obtained from the Kew Gardens Seed Information Database (https://www.kew.org/kew-gardens). Seeds stored in GBOWS were collected





from populations within the natural distribution range of the species, and dried for 1 to 6 months in a drying room where the relative humidity and temperature were maintained at 15% and 15 ℃, respectively. After drying, 50 seeds were randomly sampled from each population for five times

(sampling with replacement) and weighed the sampled seeds to the nearest 0.1 mg each time, resulting in five weights for the population. The five weights were averaged and converted to the 1000-seed weight of the population. For each species, the 1000-seed weights across all populations were further averaged and this "grand" average was used as the seed mass for the species. Seed mass variability (i.e., intraspecific variation in seed mass), ranging from zero to one, was calculated for each species as the

absolute difference between the maximum 1000-seed weight and the minimum 1000-seed weight across all the populations of the species divided by the maximum value, which is a common measure of plant trait variation (Valladares et al., 2000; Rozendaal et al., 2006). This measure is more suitable than the coefficient of variation (CV), which is sensitive to small changes in mean values when the mean is close to zero; and some plants in this study, such as orchids, have very small seed mass.

**2.2 Species distributional range size**

In this study, we estimated the distributional range size for each of the 1,426 species using ArcGIS10.2 from the global distribution of the species. Thus, the range sizes of the species were the global distribution range. Firstly, the specimen distributional information of each species was obtained from the Global Biodiversity Information Facility (GBIF.org, https://doi.org/10.15468/dl.umswqd, on 04

August 2019), the Chinese Virtual Herbarium (http://www.cvh.ac.cn/) and the Biodiversity of the Hengduan Mountains and Adjacent Areas of South-Central China websites (BHMAASCC: http://hengduan.huh.harvard.edu/fieldnotes). Specimens lacking data on GPS locations, having duplication, containing incorrect coordinates, and those taken from gardens and small oceanic islands were filtered out from our analysis. In addition, species that were cultivated, introduced, invasive, or





naturalized were also excluded from our dataset. After excluding these species records, 4,138,851 specimens of the 1,426 seed plant species were obtained. Secondly, *shapefile* (containing points) of each species was produced from the coordinates of the specimens. The *shapefile* was transformed into *raster* using the World Sinusoidal Projection at a spatial resolution of 100 km using ArcGIS10.2 (ESRI, Redlands, CA, USA). The distributional range size of each species was calculated by multiplying the number of grids the *raster* contained by 10,000 km$^2$ (100 x 100 km). In order to assess the impact of different spatial resolutions used in calculating species distributional range size, *raster* with the spatial resolution of 50 km was also used to calculate the range size. Because the distributional range size calculated at this resolution was highly correlated with the distributional range size calculated at the resolution of 100 km ($r = 0.993$, $P < 0.001$; Fig. A1), we thus only used the distributional range size calculated at the spatial resolution of 100 km in subsequent analyses.

### 2.3 Dispersal modes

Based on the published literature and floras, dispersal modes were classified to autochory (self-dispersal, e.g., by explosive seed release from fruits or gravity, $n = 223$ species), zoochory (dispersal by animals through ingestion or attachment to an animal body, $n = 468$ species), and anemochory (dispersal by wind, $n = 735$ species) according to the morphological features of their seeds or fruits (Pérez-Harguindeguy et al., 2013). For example, seeds or fruits with wings, hairs or pappus were considered wind dispersed; seeds or fruits with an aril or flesh offering a succulent reward for consumers were classified as zoochory; and seeds or fruits lacking modifications pertaining to the other two categories were classed as autochory (unassisted dispersal) (Qi et al., 2014).

### 2.4 Construction of phylogenetic tree and statistical analyses

For all the species used in our analysis, the scientific names were checked and standardized according to



the Plant List (http://www.theplantlist.org/). Different varieties and subspecies of a given species were considered to belong to the same species. The phylogenetic tree was extracted from a previously published supertree using the 'phylo.maker' function in R package *V.PhyloMaker* (Jin and Qian, 2019),

which was based on the APG classification of flowering plants (Zanne et al., 2014). The 'multi2di' function in the *ape* package was used to randomly resolve polytomies in the phylogenetic tree. To test the phylogenetic signal in species distribution, 'phylosig' function in the R package *phytools* was used to calculate Pagel's λ, which is ranged between 0 and 1. λ = 0 means that the evolution of the trait is phylogenetically independent, and λ = 1 indicates that trait evolution follows the Brownian motion. Any

value of λ significantly higher than zero is regarded to have a phylogenetic signal approaching Brownian motion to a different degree (Arène et al., 2017).

Because closely related species tend to have similar traits, interspecific analyses can be compromised by phylogenetic relatedness (Felsenstein, 1985; Lynch, 1991). In our case, species' range size is not phylogenetically independent. We thus used a phylogenetic generalized least squares (PGLS)

regression to determine the effects of seed mass (SM), intraspecific variation in seed mass (ISM) and dispersal mode (DM) on the distributional range size (RS) of species (Swenson, 2014). The SM×DM and ISM×DM interaction terms were also included in the PGLS model, in order to show effects of SM and ISM on distributional range size among dispersal modes. The regression model was RS = $\beta_0$ + $\beta_1$SM + $\beta_2$ISM + $\beta_3$DM + $\beta_4$SM×DM + $\beta_5$ISM×DM. The PGLS was implemented using 'gls' function

in *nlme* package, and the possible phylogenetic dependence in species' range size was incorporated in a form of a phylogenetic variance-covariance matrix in gls.

We further used 'varpart' function in *vegan* package to partition the variances in range size explained by seed mass, seed mass variability, dispersal mode, and genus (regarded as phylogeny). Because our phylogenetic tree had some polytomies at the species-level, genera were used as a

surrogate in the phylogeny. Variation partitioning is a linear model, which does not require the type of





explanatory variables, and hence is suitable to our data structure (Borcard et al., 2018).

In the analyses of this study, the values of species range size and seed mass were $\log_e$-transformed to reduce data skewness and downplay extreme values; and the log-transformed seed mass and seed mass variability were standardized to make their coefficients (i.e., effect size) comparable. Seed mass and seed mass variability were each standardized by subtracting the smallest value across all 1426 species and divided by the difference between the largest value and the smallest value. All statistical analyses in this study were conducted using R4.0.2 (R Core Team, 2020).

## 3 Results

### 3.1 Effects of phylogeny on species distributional range size

We detected a strong phylogenetic signal in species distributional range size for the study species ($\lambda = 0.627$, $P < 0.001$), with the signal being stronger in gymnosperms ($\lambda = 0.975$, $P < 0.05$) than in angiosperms ($\lambda = 0.423$, $P < 0.001$). The phylogenetically closely related species had more similar range size than that for distantly related species.

### 3.2 Effects of seed traits on species distributional range size

The results of the phylogenetic generalized least squares regression showed that seed mass had a negatively strong association with species distributional range size (effect size = -13.974, $P < 0.001$; Fig. 1, Table A1), while the effect of seed mass variability on species distributional range size was not significant (effect size = 0.459, $P = 0.109$). Dispersal mode was also significantly associated with species' range size. In the PGLS model, autochorous (explosive/gravity dispersal) species was treated as the baseline dispersal mode. Compared to zoochory (dispersal by animal ingestion or attachment to an animal body) and anemochory (dispersal by wind), autochorous species had significantly larger range



size after the effects of seed mass and seed mass variability were accounted in the interaction terms between seed traits and dispersal modes (Fig. 1, Table A1). The interaction terms between seed mass/seed mass variability and dispersal mode (i.e., seed mass × anemochory, seed mass × zoochory and seed mass variability × zoochory) were significantly positive (effect size = 7.527, $P < 0.001$; effect size = 12.637, $P < 0.001$; effect size = 1.824, $P < 0.001$ respectively), indicating the distributional range sizes of anemochorous and zoochorous species were strongly subject to seed mass and its intraspecific variation (Fig. 1, Table A1).

### 3.3 Joint effects of seed traits and phylogeny on species' range size

Variation partitioning showed that the effects of seed mass, seed mass variability, dispersal mode and phylogeny together explained 46.82% of the variance of species' range size (Fig. 2). Of the explained variation, seed mass (including mass variability) contributed a pure 11.38% fraction, phylogeny contributed a pure 21.31%, and a small fraction from the pure dispersal mode (0.72%). We also noted a considerable joint effect of seed traits and phylogeny (13.41%) on species' range size (Fig. 2).

### 4 Discussion

### 4.1 The relationship between phylogeny and species distributional range size

We found a significant phylogenetic signal associated with species distributional range size. This result suggests that closely related species are more similar in distribution range size than distantly related species. It corroborates some studies (e.g., Hunt et al., 2005; Martin and Husband, 2009), but does not support those of Webb and Gaston (2003) which showed the distributional range sizes of closely related species were not more similar to each other than expected by chance. This discrepancy may be due to the different evolutionary history of the studied taxa as well as the heritability of their life-history traits,





which can play a critical role in the establishment and persistence of species, and thus influence their distributional range sizes (Angert and Schemske, 2005; Umaña et al., 2018). It is worth noting that

Webb and Gaston (2003) studied birds that have much stronger dispersal ability than seed plants, which may contribute to the difference between our studies. Seed traits associated with range size can also change over evolutionary time, which in turn could alter the range size of a species' distribution (Blomberg et al., 2003). Furthermore, the geographic distribution range of a species can be influenced by its ecological tolerances associated with life-history traits (Geber and Griffen, 2003; Latimer and

Zuckerberg, 2021). Our results imply that the geographic distribution of related species may have a similar response to patterns of climate change at a regional scale, due in part, to phylogenetic constraints on the distributional range of species. Here, it seems likely that closely related species have commonly evolved seed traits that result in shared adaptative strategies to climate change, although this causal mechanism requires further empirical study in the field.

**4.2 Effects of seed traits on the distribution of species**

We found a very strong negative relationship between seed mass and species range size, meaning larger seeds having smaller range size (Fig. 1, Table A1). This result is consistent with previous studies that also found a significant relationship between seed mass and range size (Morin and Chuine, 2006; Proches et al., 2012). Different from the effect of seed mass, seed mass variability had no or a weak

positive association with distributional range size.

The PGLS model showed that the range sizes of zoochorous (animal-dispersed) and anemochorous (wind-dispersed) species were significantly smaller than that of autochorous (explosive/gravity dispersed) species (Fig. 1). This may appear counterintuitive at the first glance but was resulted after the effects of the interactions between seed mass (and mass variability) and dispersal mode were taken

accounted. These strong positive interaction terms (except the interaction between seed mass variability



and wind dispersal) shown in Fig. 1 indicate that the range sizes of species with different dispersal modes are strongly subject to seed mass (and also mass variability). For example, zoochorous species with large seed mass and mass variability have significantly larger range size than species that have similar seed traits but dispersed by explosive gravity. This dependence of species distributional range size on the interactions between seed mass and dispersal mode is further confirmed by a simpler PGLS model that excludes all the interactive terms between seed mass (and mass variability) and dispersal mode. The results of this model in Appendix Table A2 show that zoochorous species had significantly larger range size than that of autochorous and anemochorous species ($P < 0.001$), while the latter two groups were not significantly different ($P = 0.257$).

Although intraspecific seed mass variability did not seem to affect distributional range size of autochorous and anemochorous species, the variability was strongly positively associated with range size of zoochorous species. This may be because species with large variation in seed mass could have greater colonization ability in various habitats and seeds of zoochorous species with long dispersal distance have more chances to arrive at heterogeneous habitats than seeds of autochorous and anemochorous species. Given that small- and large-seeded species are shown to adapt to different habitats (Silvertown, 1989), it seems likely that zoochorous species may experience trade-offs between competition ability and dispersal ability through seed mass variation (Chen et al., 2018), resulting in a similar effect for seed mass on species distributional range size at the geographic scale.

It is interesting to note that Sides and Sloat (2014) found that species with greater intraspecific variation in specific leaf area (SLA) have wider ecological breadth. Due to its potential role in modulating the response of plant species to environmental changes, greater intraspecific functional variability enables species to adjust to a wider range of competitive and abiotic conditions (Sides and Sloat, 2014; Basnett and Devy, 2021). Plastic responses of seed mass to heterogeneous environments may be related to molecular signals at a single gene or of the entire genome (Nicotra et al., 2010) and



thus influence the distributional range size of species (Savolainen et al., 2007). Distributional patterns of plant species may reflect the fact that individuals within a species have different levels of genetic variation in association with seed mass, thus facilitating the species to adapt to a broad spectrum of environments (Völler et al., 2012).

## 4.3 Effects of seed mass, seed dispersal and phylogeny on species' range size

Our results show that seed traits and phylogeny jointly affect species distributional range size, indicating that species distribution may be limited by ecological and evolutionary processes (Fig. 2). There are two possible reasons for this relationship: (1) the evolution of both seed mass and dispersal mode is phylogenetically conserved (Gallagher and Leishman, 2012; Chen et al., 2018; Kang et al., 2021); and (2) seed mass and seed dispersal mode are not evolutionarily independent but are
constrained by evolutionary history, e.g., phylogenetic divergences in dispersal syndrome is related to divergences in seed mass (Moles et al., 2005). However, we also need to recognize that more than 50% of the variance in species distribution in our study remains unexplained. This result suggests that climatic tolerance, competition, colonization ability and other geographic factors could also be important for affecting species distribution (Morin and Chuine, 2006).

## 5 Conclusions

This study provides evidence that seed mass, intraspecific seed mass variation, seed dispersal mode and phylogeny contribute to explaining species distribution variation on the geographic scale. We found that (1) species distributional range size was significantly constrained by phylogeny, seed mass and its intraspecific variability, and seed dispersal mode; (2) the effects of seed mass and seed mass variability
on species distribution varied among dispersal modes; and (3) seed mass, dispersal mode and phylogeny together explained 46.82% of the variance associated with species distributional range size. Despite that





more than half of the variation in species distribution is left unexplained, our study clearly shows the importance of including seed life-history traits in modeling and predicting the impact of climate change on species distribution of seed plants.


*Data availability.* The data are available from the freely accessible databases cited in the manuscript.

*Authors contribution.* DZL, LMG and FH designed the study; KC and XYY collected data; KC conducted statistical analysis and generated the graphs; KC, KSB and LMG wrote the manuscript; DZL, FH and XYY revised the manuscript. All authors reviewed and approved the final manuscript.

*Competing interests.* All authors have no conflict of interest.

*Acknowledgements.* We are grateful to Xie He, Jie Cai, Ting Zhang, Jian-Jun Jin, Hua-Jie He, Tuo-Jing Li and other staff of the Germplasm Bank of Wild Species for assistance in seed mass data or specimen collection data. We also thank Ming-Cheng Wang for helping calculate species range size.

*Financial support.* This study was supported by the Strategic Priority Research Program of Chinese

Academy of Sciences (XDB31000000), the International Partnership Program of Chinese Academy of Sciences (151853KYSB20190027), the National Natural Science Foundation of China (32160078), and the National Key Basic Research Program of China (2014CB954100).

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





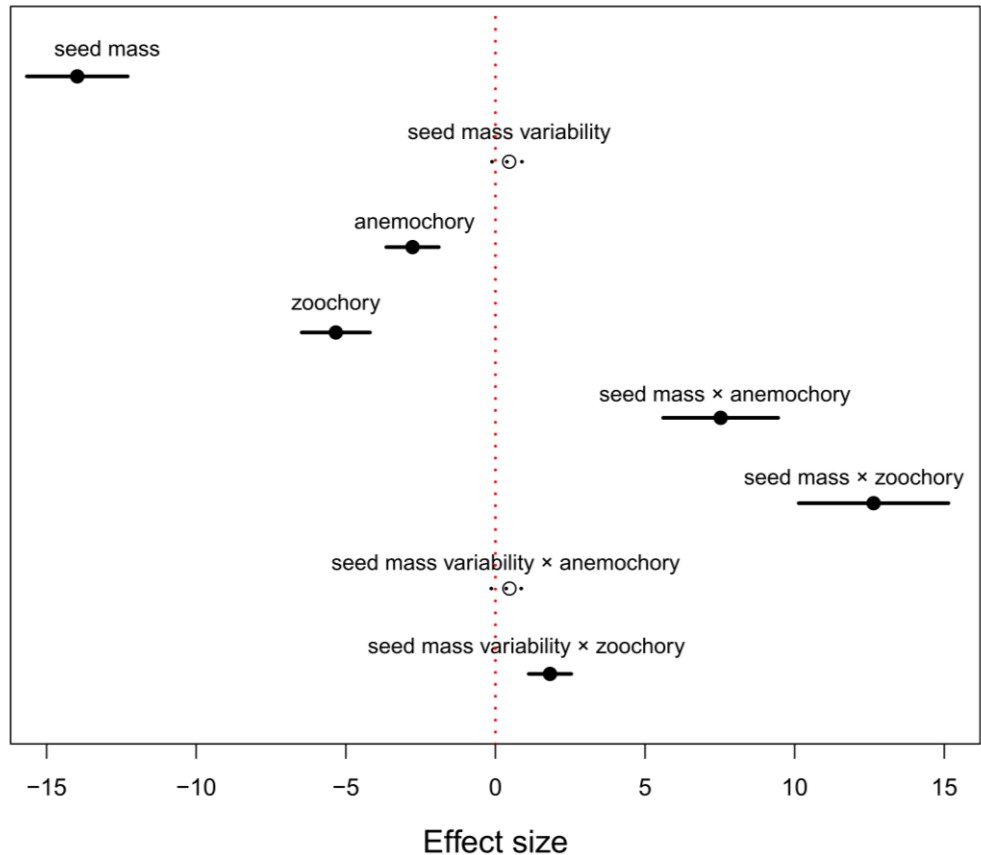

Figure 1. Effects of seed mass and seed mass variability on species distributional range size in autochorous, zoochorous and anemochorous species. In the PGLS model, autochory was treated as a baseline dispersal mode. The black segments represent the effect sizes are statistically significantly different from 0 ($P < 0.05$), while the pointed lines with open circle indicate non-significant effect sizes.





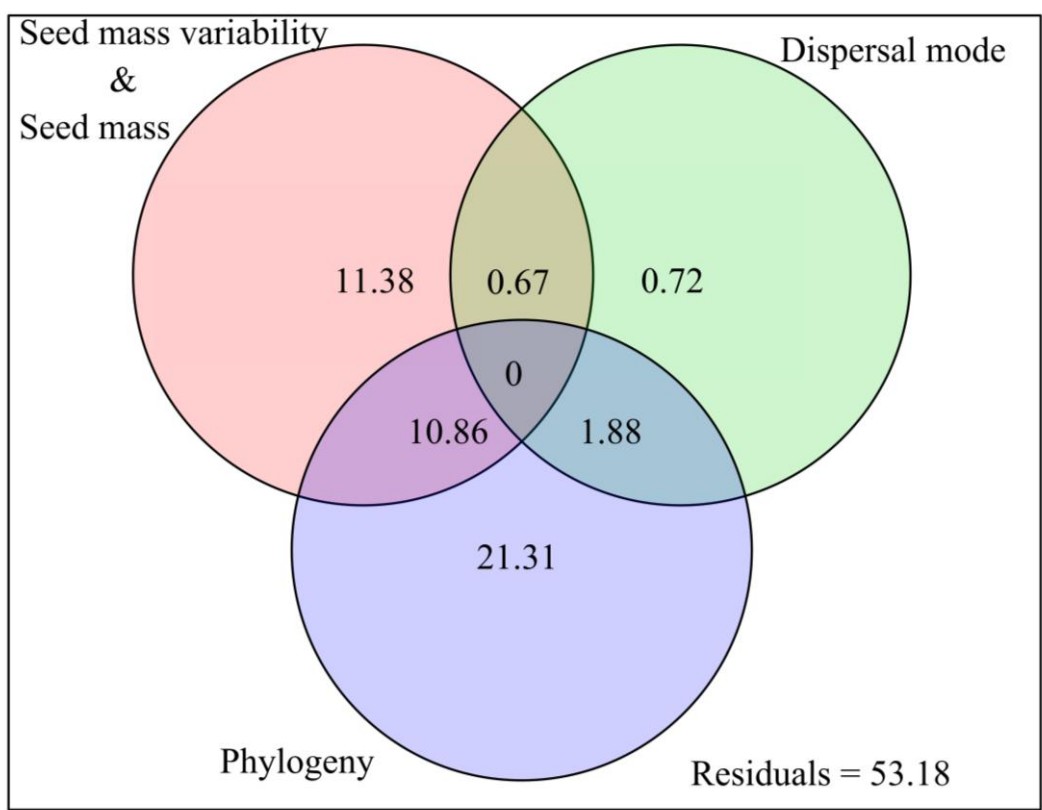

Figure 2. Variation partitioning of seed mass, seed mass variability, dispersal mode, and phylogeny for species distributional range size.





## APPENDICES

Table A1. The phylogenetic generalized least squares regression for modeling the effects of seed mass, seed mass variability, dispersal mode, seed mass × dispersal mode and seed mass variability × dispersal mode interaction terms on species distributional range size. The graphic presentation of the results of this table is given in Figure 1 in the main text.

| Variable | Effect size±SE | $t$-value | $P$-value |
|---|---|---|---|
| Intercept | 18.406±5.612 | 3.279 | 0.001 |
| Seed mass | -13.974±0.842 | -16.593 | <0.001 |
| Seed mass variability | 0.459±0.286 | 1.604 | 0.109 |
| Anemochory | -2.769±0.438 | -6.318 | <0.001 |
| Zoochory | -5.333±0.570 | -9.358 | <0.001 |
| Seed mass×anemochory | 7.527±0.960 | 7.838 | <0.001 |
| Seed mass×zoochory | 12.637±1.250 | 10.105 | <0.001 |
| Seed mass variability×anemochory | 0.468±0.303 | 1.545 | 0.123 |
| Seed mass variability×zoochory | 1.824±0.355 | 5.140 | <0.001 |



Table A2. The phylogenetic generalized least squares regression for modeling the effects of seed mass, seed mass variability and dispersal mode, without interaction terms, on species distributional range size. In the model, autochory (explosive/gravity dispersal) was treated as the baseline dispersal mode. The results in the table show zoochorous species had significantly larger range size than that of autochorous species ($P < 0.001$), while the range size of anemochorous (wind dispersal) species and that of autochorous species were similar ($P = 0.257$).

| Variable | Effect size±SE | $t$-value | $P$-value |
|---|---|---|---|
| Intercept | 16.018±5.988 | 2.675 | 0.008 |
| Seed mass | -7.424±0.422 | -17.611 | <0.001 |
| Seed mass variability | 1.1±0.092 | 11.974 | <0.001 |
| Anemochory | 0.323±0.285 | 1.133 | 0.257 |
| Zoochory | 1.16±0.295 | 3.928 | <0.001 |



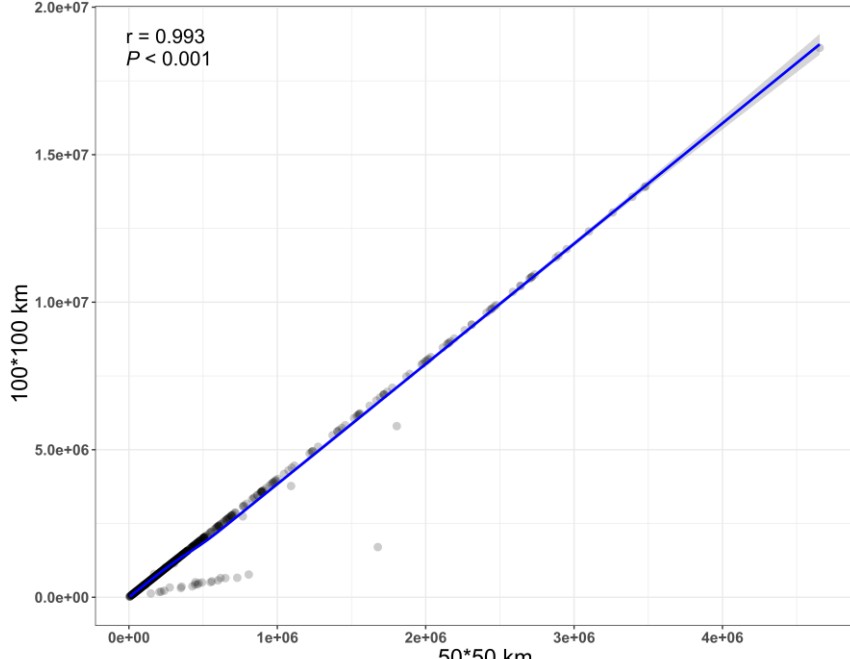

Fig. A1 Relationship between distributional range size calculated at the spatial resolution of 50 km and the range size calculated at the spatial resolution of 100 km.