# Peer review of "Seed traits and phylogeny explain plant's geographic distributions"

_EGUsphere, 2022_

## Author Response (AR1)

**Associate Editor's Comments to Authors**

Associate Editor decision: Publish subject to minor revisions (review by editor)

Comments to the author:
  Dear Authors,
   Both referees recommend to accept your manuscript without further revisions. Please go ahead an upload the final version of your manuscript so that we can move on with the publication. Thank you & all the best,
   Anja Rammi

**Response**: Thank you.

**Reviewers' Comments to Authors**
**Reviewer #1**
The manuscript by Kai Chen et al presents an interesting study about the relationship between seed traits, phylogeny and plant distribution. The authors quantify the joint effects of key seed traits and phylogeny on species distribution. They found that seed mass and its intraspecific variation were also important in limiting species distribution, but their effects were different among species with different dispersal modes. I think the information provided here is relevant for plant geography, as it shows that seed mass, seed mass variability, seed dispersal mode and phylogeny together explained 46.82% of the variance in species range size. This finding underscores the necessity to include seed traits and the phylogenetic history of species in climate-based niche models for predicting the response of plant geographic distribution to climate change.

The manuscript contains a lot of data and analysis, and I thought it would be hard to read, but it is easy to read, because it is short, clear and punchy.

**Reviewer #2**
Chen et al.'s manuscript "Seed traits and phylogeny explain plant's geographic distributions" quantified the joint effects of key seed traits and phylogeny on species' distribution based on a large-scale sampling of 1426 seed plants representing 501 genera of 122 families, using 4,138,851 specimens to model species distributional range size. The result showed that species distributional range was significantly constrained by phylogeny, and seed mass and its intraspecific variation, but their effects varied among species with different dispersal modes. Seed mass variability, seed dispersal mode and phylogeny together explained nearly half of the variance in species range size. This study highlights the necessity to include seed traits and the evolutionary history of species in niche models for predicting the response of plant geographic distribution to the climate change. These findings will improve our understanding on the mechanisms of shaping the geographic distribution of plant species.

The resubmitted manuscript was much improved, and all my concerns for the first review were addressed and resolved. The text is well written. The language is appropriate. The discussion and conclusions seem fairly well supported by their results. I think the manuscript would potentially deserve to be published.

**Response**: Thank you for the assessment.